

# Biomarkers associated with cell-in-cell structure in kidney renal clear cell carcinoma based on transcriptome sequencing

Zehua Wang[1] and Zhongxiao Zhang[2]

[1] Department of Urology, Qilu Hospital, Shandong University, Jinan, China
[2] Department of Urology, Qilu Hospital (Qingdao), Shandong University, Qingdao, China

Corresponding author
Zhongxiao Zhang,
a18561810507@163.com

## ABSTRACT

**Background**. Kidney renal clear cell carcinoma (KIRC), the main histological subtype of renal cell carcinoma, has a high incidence globally. Cell-in-cell structures (CICs), as a cellular biological phenomenon, play pivotal roles in cell competition, immune evasion and tumor progression in the context of KIRC.

**Methods**. Data for this study were sourced from The Cancer Genome Atlas (TCGA), International Cancer Genome Consortium (ICGC), and Gene Expression Omnibus (GEO) databases. Differentially expressed genes (DEGs) were identified using the limma package. Enrichment analyses were performed using the clusterProfiler package. Support vector machine-recursive feature elimination (SVM-RFE) and Least Absolute Shrinkage and Selection Operator (LASSO) regression, implemented *via* the caret and glmnet packages in R, were used to select biomarkers. The accuracy of these biomarkers was verified by using the receiver operating characteristic (ROC) curve as well as *in vitro* experiments (CCK-8 assay, wound healing assay, Transwell assay, and quantitative real-time PCR). The CIBERSORT algorithm was applied to explore the association between immune infiltration and the biomarkers. Further analysis explored the association between these biomarkers and clinicopathological characteristics of KIRC. For single-cell data, the Seurat package is used to read the sample data, and the SCTransform function is employed for normalization.

**Results**. This study identified 1,256 DEGs which enriched in T-cell immune system regulation processes. Five hub genes (*CDKN2A*, *VIM*, *TGFB1*, *CTSS*, and *CDC20*) were biomarkers with area under the curve (AUC) values > 0.8, indicating high predictive performance. *In vitro* validation experiments demonstrated that the expressions of all five biomarkers in KIRC cells were elevated, and the knockdown of *CTSS* could inhibit the migration and invasion of KIRC cells. Immune infiltration analysis showed higher proportions of T-cells and macrophages in tumor tissues. *CDKN2A* and *CDC20* expressions correlated significantly with stage and grade, while *TGFB1*, *CDKN2A*, and *CDC20* were highly expressed in proliferative tumor cells.

**Conclusion**. This study provides new biomarkers for KIRC, offering valuable insights into its developmental mechanisms for the research of CIC in this disease.

## INTRODUCTION

Renal cell carcinoma (RCC) is the most frequent malignancy of the kidney (*Makhov et al., 2018a*). Globally, there are approximately 434,419 new cases each year, accounting for about 2.2% of all cancers and ranking 14th (*Bray et al., 2024*; *Tian et al., 2024*). Its mortality rate is approximately 1.6%, placing it 16th among all cancers (*Bray et al., 2024*). Smoking, overweight, and obesity are established risk factors for kidney cancer (*Frew & Moch, 2015*). Kidney renal clear cell carcinoma (KIRC) is currently the main histological subtype of kidney cancer, accounting for about 80–90% of kidney cancer patients, but it has a poor prognosis (*Wang et al., 2019*). Studies have reported that KIRC exhibits heterogeneity in clinical pathology, molecular, and cellular aspects (*Xie et al., 2020*). However, due to the limited availability of biomarkers for early detection and prognosis prediction, the prognosis for KIRC patients is generally poor (*Cui et al., 2020*). Hence, it is urgent to investigate the pathogenesis of KIRC and explore new molecular biomarkers for diagnosis and prognosis (*Seyfinejad & Jouyban, 2022*; *Liu et al., 2024*; *Zhang et al., 2023*).

Cell-in-cell structures (CICs) represent a cellular biological phenomenon where one living cell is enclosed within another living cell (*Wang et al., 2020*). CICs were initially discovered in the context of tumor biology and are considered to play pivotal roles in cell competition, immune evasion, and tumor progression (*Fais & Overholtzer, 2018*; *Huang, Chen & Sun, 2015*). CICs exert various effects on cellular behavior and the functions of both external and internal cells, encompassing cell death, cell proliferation, and immune regulation (*Wang, 2015*). Increasing evidence suggests that CICs may hold prognostic and diagnostic value for cancer patients (*Su et al., 2022*; *Chen et al., 2013*). It has been proposed that tumor cells may utilize CIC-mediated internal cell death as a means of immune evasion (*Sun & Chen, 2022*). Consequently, research on CICs in renal cancer is of great significance for understanding tumor biology and developing novel therapeutic strategies.

Based on the above background, this study aimed to mine molecular biomarkers for the diagnosis and prognosis of KIRC based on CICs-related genes, and establish a corresponding diagnostic model. Additionally, we conducted in-depth analyses of the correlation between these biomarkers and immune infiltration, their relationship with the clinicopathological characteristics of KIRC, and their distribution and expression in the kidney. The significance of this study lies not only in providing new scientific evidence for early diagnosis, prognosis assessment, and personalized treatment of KIRC, but also in offering new possibilities for exploring the pathogenesis of renal cancer and developing novel therapeutic strategies.

## MATERIALS AND METHODS

### Data acquisition

The current research encompasses datasets from four aspects. Firstly, RNA-Seq data for The Cancer Genome Atlas (TCGA)-KIRC were downloaded using the TCGA Genomic Data Commons (GDC) application programming interface (API). The FPKM values were converted to TPM and then log2-transformed, with a total of 513 primary tumor samples and 72 adjacent normal control samples retained. Secondly, expression profiles for The Renal Cell Cancer-European Union (RECA-EU)/Renal cell carcinoma were obtained from the International Cancer Genome Consortium (ICGC) database, encompassing 91 primary tumor samples and 45 adjacent normal control samples. Furthermore, the KIRC single-cell dataset GSE224630 was obtained from the Gene Expression Omnibus (GEO) database. The GSE224630 dataset includes tumor samples from 6 patients with untreated clear cell renal cell carcinoma. Finally, 101 CIC-correlated genes were sourced from previous literature (*Song et al., 2022*; *Ren et al., 2024*).

### Identification and enrichment analysis of DEGs

The limma package (*Ritchie et al., 2015*) was utilized in the TCGA-KIRC and ICGC datasets to screen for DEGs between KIRC patients and normal controls, and to identify common upregulated genes across both datasets (*Song et al., 2023a*). The criteria for selecting DEGs were a $p < 0.05$ and $|\log2FC| > 1$. Additionally, the clusterProfiler package in R (*Yu et al., 2012*) was employed to enrich the Gene Ontology (GO) and Kyoto Encyclopedia of Genes and Genomes (KEGG) of these DEGs. Adjusted $p < 0.05$ indicated significantly enriched pathways (*Song et al., 2023b*).

### Machine learning for key gene selection, diagnostic model development, and validation

The overlapping genes obtained from the intersection of DEGs and CIC genes were further screened using machine learning techniques. The rfe function from the R package caret (*Kuhn, 2008*) was employed, utilizing the svmlinear method. During the selection of the number of features through recursive feature elimination (RFE), the cross-validation (CV) accuracy of the support vector machine (SVM) model was used to screen for key disease genes. Feature selection was also conducted using Least Absolute Shrinkage and Selection Operator (LASSO) regression from the glmnet package (*Friedman et al., 2021*), with key parameters set to nfolds = 10 and family = 'binomial'. Finally, the feature genes selected by both the SVM-RFE and LASSO methods were intersected to obtain the biomarkers for this study. Subsequently, the R package e1071 (*Meyer et al., 2019*) was used to construct a diagnostic model using the SVM method. The accuracy of the biomarkers selected by machine learning was tested by plotting receiver operating characteristic (ROC) curves for KIRC samples and normal controls in the TCGA-KIRC training set. A larger the area under the curve (AUC) indicated a higher accuracy of considering the genes as hub genes. The validity of these genes was further verified using the ICGC validation set using the same method.

### Correlation analysis of biomarkers with immune infiltration

The CIBERSORT package (*Newman et al., 2015*) was employed to quantify different immune cells in KIRC samples and control samples. Spearman correlation coefficients were used to perform correlation analyses between biomarkers and immune cells.

### Processing and analysis of scRNA-seq data from KIRC

The Read10X function from the Seurat package (*Hao et al., 2024*) was utilized to read the scRNA-seq data for each sample in GSE224630, retaining cells with a gene count between 200 and 5000 and a mitochondrial gene proportion of less than 10%. Subsequently, the SCTransform function (*Hao et al., 2024*) was applied for normalization. After Principal Component Analysis (PCA) dimensionality reduction, the harmony package (*Korsunsky et al., 2019*) was used to remove batch effects among different samples. The RunTSNE function (*Hao et al., 2024*) was then employed for t-Distributed Stochastic Neighbor Embedding (TSNE) dimensionality reduction. Finally, the FindNeighbors and FindClusters functions (*Hao et al., 2024*) were used for clustering, with parameters set to dims = 1:25 and resolution = 0.1. Cell subpopulations were annotated based on marker genes provided by the CellMarker2.0 database (*Hu et al., 2022*). The expression patterns of the biomarkers obtained in this study across different cell types were subsequently investigated.

### Cell culture and transfection

Two cell lines, human embryonic kidney 293T (CRL-3216) and human renal clear cell adenocarcinoma cell 786-O (CRL-1932), were all obtained from the American Type Culture Collection (Manassas, MD, USA). These cells were cultured in DMEM (11965092, Gibco, Waltham, MA, USA) or RPMI 1640 medium (11875093, Gibco, Waltham, MA, USA) with the supplementation of 10% fetal bovine serum (FBS) (S9020, Solarbio Lifesciences, Beijing, China) and 1% penicillin-streptomycin (15140148, Gibco, Waltham, MA, USA). All cells were tested *via* short tandem repeat profiling and incubated in the incubator at 37 °C with 5% $CO_2$.

For the liposome transfection, the small interfering RNA against *CTSS* (si-*CTSS*) and the control small interfering RNA (si-NC) were all purchased from GenePharma (Shanghai, China) and transfected into 786-O cells with the use of lipofectamine 2000 transfection reagent (11668027, Invitrogen, Carlsbad, CA, USA) as per the manuals. The sequences applied for the transfection were 5′-TCACATATAAGTCAAACCCTA-3′.

### CCK8 assay

During the logarithmic phase, 786-O cells were plated in a 96-well plate with a density of $1 \times 10^4$ cells per well and incubated at 37 °C in an atmosphere containing 5% $CO_2$ for 0, 24, or 48 h. Subsequently, 10 μL of CCK-8 reagent was added to the medium, and the samples were incubated at 37 °C for 2 h. To generate the CCK-8 curve, the absorbance was measured at 450 nm, which served as the ordinate, while time was represented on the abscissa. The results were averaged from three independent experimental repeats.

### Cell migration assay

The transfected 786-O cells ($5 \times 10^5$ cells/well) were cultivated in a six-well plate with serum-free media. When they achieved complete confluence, a 200-μL sterile pipette tip

was used to create an artificial scratch on the monolayer. Forty-eight h later, the cells were photographed by an inverted optical microscope (DP27, Olympus, Tokyo, Japan), and the wound closure (%) was quantified correspondingly to assess the migration of KIRC cells (*Zhang et al., 2024*). Wound closure (%) = (Initial scratch width–scratch width at measurement time point)/Initial scratch width ×100%.

## Cell invasion assay

For the invasion assay, 786-O cells ($1 \times 10^5$/100 µL) were suspended in 200 µL serum-free medium and plated in the upper Transwell chamber (3422, Corning, Inc., Corning, NY, USA) coated with matrix gel (C0372, Beyotime, China), while the lower chamber was filled with 700 µL culture media containing 10% bovine calf serum. After 48 h, the invaded cells were fixed by 4% paraformaldehyde (P0099, Beyotime, Shanghai, China) and stained with 0.1% crystal violet (C0121, Beyotime, Shanghai, China) for 30 min. Then, three random fields were observed under an inverted optical microscope (DP27, Olympus, Japan), and the number of invaded cells was quantified (*Wang et al., 2023*).

## QRT-PCR experiment

Following the instructions, total RNA was isolated from 293T and 786-O cells using the TriZol total RNA extraction kit (15596026, Invitrogen, Carlsbad, CA, USA). Subsequently, the concentration of the isolated RNA was determined. Then, complementary DNA was synthesized by reverse transcription with a relevant assay kit (D7178S, Beyotime, Shanghai, China). After that, SYBR Green qPCR Mix (D7260, Beyotime, Shanghai, China) was used for the PCR assay according to the protocols. Finally, the relative level was calculated by the $2^{-\Delta\Delta CT}$ method with *GAPDH* as the reference gene. The qRT-PCR primers used in this study were designed according to National Center for Biotechnology Information (NCBI) sequences using Primer Premier 6 software. The sequences of the primers used were presented in Table S1.

## Statistical analysis

All statistical analysis was conducted using R (version 3.6.0). The *t*-test was performed on continuous variables between two groups. Data normality was assessed using the Shapiro–Wilk test. If normality assumptions were violated, the Wilcoxon rank-sum test was performed as a supplementary analysis. To investigate the correlation between gene expression and immune cell fractions, Spearman's rank correlation test was employed. A $p < 0.05$ signified a significant level. Sangerbox (http://sangerbox.com/) provided assistance with this study (*Shen et al., 2022*).

# RESULTS

## Differential gene selection and enrichment analysis

Comprehensive differential gene expression analysis was conducted on tumor and control samples within the TCGA-KIRC and ICGC datasets (Figs. 1A–1B). Subsequently, upregulated genes common to both the TCGA and ICGC datasets were identified, totaling 1,256 (Fig. 1C). Functional enrichment analysis, including KEGG pathway analysis and GO

analysis, was performed on these commonly upregulated genes. The top 10 enriched KEGG pathways included Human T-cell leukemia virus 1 infection, PI3K-Akt signaling pathway, Cytokine-cytokine receptor interaction, Phagosome, Epstein-Barr virus infection, and cell adhesion molecules (CAMs) among others (Fig. 1D). The results of the GO functional enrichment analysis covered biological process (BP), cellular component (CC), and molecular function (MF) aspects. The top five GO functional enrichments indicated that these genes were significantly involved in critical processes related to immune system regulation, such as regulation of lymphocyte activation, MHC protein complex, regulation of T cell activation, peptide antigen binding, T cell activation, regulation of leukocyte activation, and leukocyte cell–cell adhesion (Fig. 1E).

## Machine learning for biomarkers screening

Twelve overlapping genes were obtained by intersecting the DEGs with CIC genes. When selecting the number of features using the RFE method, the trend of the CV accuracy of the SVM model with the number of features can be observed. It can be seen that when the number of selected features is six, the model's CV accuracy reaches its maximum (Fig. 2A). To further screen the genes, LASSO regression analysis was employed. Figure 2B displayed the changes in regression coefficients for different gene features in LASSO regression as the penalty parameter ($\lambda$) varies, showing the deviance at the optimal $\lambda$ value determined by 10-fold CV. By intersecting the genes obtained from both the RFE-SVM and LASSO methods, five hub genes were ultimately identified as biomarkers in this study, namely *CDKN2A*, *VIM*, *TGFB1*, *CTSS*, and *CDC20* (Fig. 2C). Figures 2D–2E demonstrated the differential expression of these hub genes between tumor samples and normal controls, showing that all biomarkers are expressed higher in the tumor group than the control group in both the TCGA training set and the ICGC validation set ($p < 0.0001$).

## Construction and verification of the diagnostic model

The five biomarkers were integrated to establish a diagnostic model, and the predictive ability of this model in the TCGA training set was assessed using ROC curves. Each curve represents a hub gene, and the AUC value was used to explore the gene's ability to distinguish between diseases. The results showed that the AUC > 0.8 (Fig. 3A), indicating that these genes had high predictive power. Figure 3B displayed the ROC curve for the overall predictive ability of the constructed SVM model, with an AUC of 0.962, suggesting that the model exhibits very high predictive performance in distinguishing between KIRC and normal samples. The confusion matrix of the classification model demonstrated the classification results for the tumor group and the control group (Fig. 3C), further validating the robustness of the model in distinguishing between the two groups of samples. Subsequently, the same method was used for validation in the ICGC validation set. Both the AUC values for individual genes and the overall AUC value for the SVM model were above 0.8, indicating that the diagnostic model in this study has high predictive performance (Figs. 3D–3F).

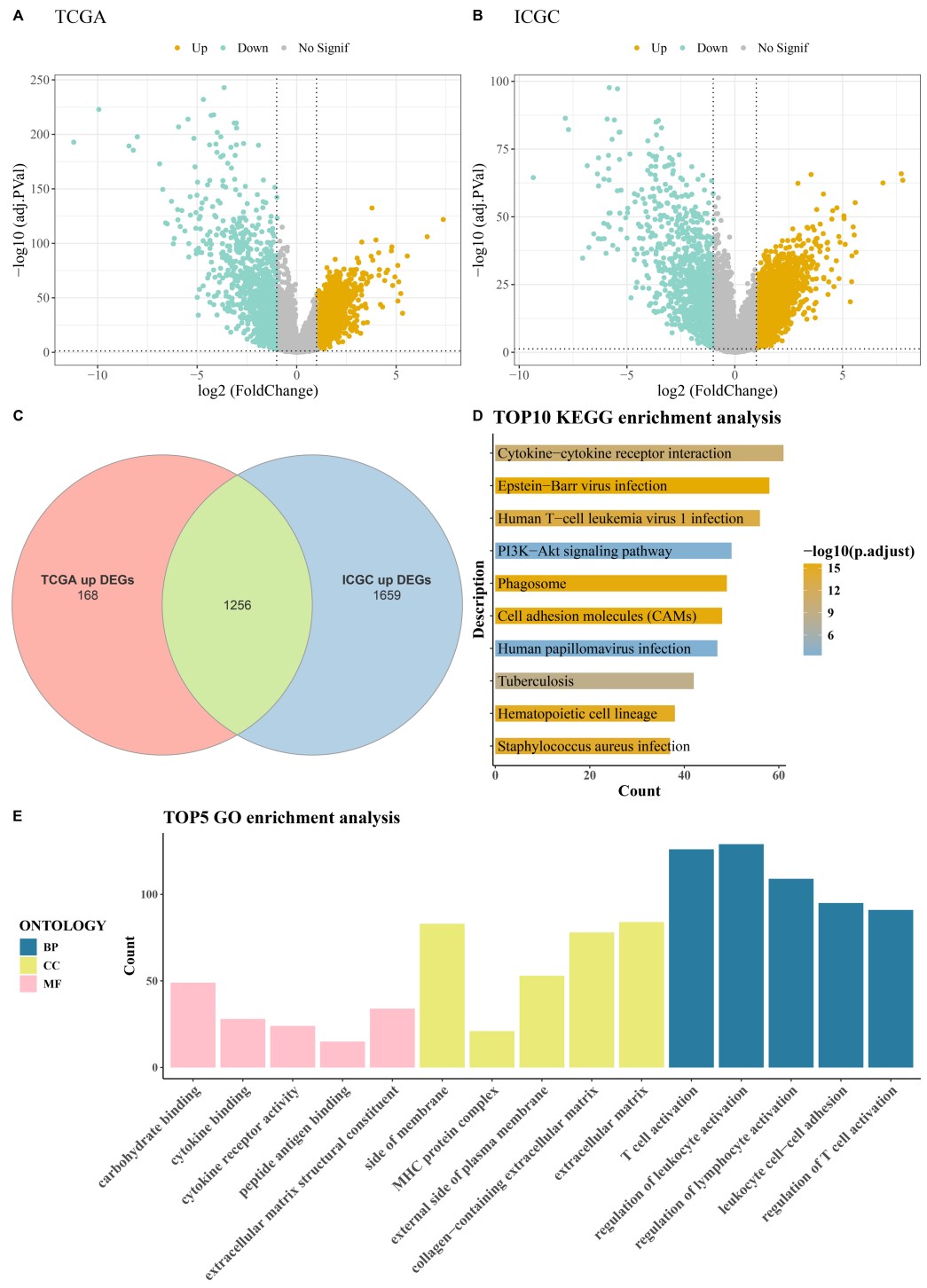

**Figure 1** **Acquisition and enrichment analysis of differential genes.** (A) Volcano plot of DEGs in the TCGA cohort; (B) Volcano plot of DEGs in the ICGC cohort; (C) Venn diagram of upregulated genes common to both the TCGA and ICGC cohorts; (D) KEGG pathway enrichment analysis of DEGs; (E) GO enrichment analysis of DEGs in terms of BP, CC, and MF.

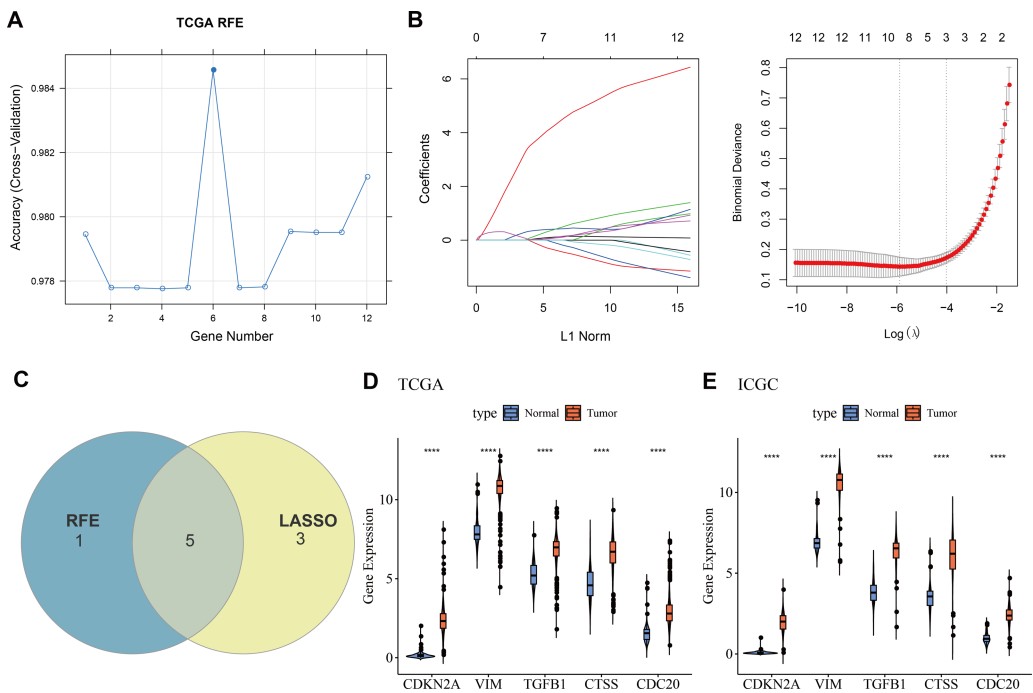

**Figure 2** **Machine learning for biomarkers screening.** (A) The curve of CV accuracy varying with the number of features selected by the RFE method; (B) The changes in regression coefficients of gene features in the LASSO regression model and the optimal penalty parameter (λ) determined through CV. The red dashed line indicates the selected optimal λ value, which corresponds to a relatively small number of features while ensuring good predictive performance of the model; (C) A Venn diagram showing the intersection of feature genes selected by both the RFE-SVM and LASSO methods; (D) The expression levels of feature genes in the tumor group *versus* the control group within the TCGA training set, with **** indicating $p < 0.0001$; (E) The expression levels of feature genes in the tumor group *versus* the control group within the ICGC validation set, with **** indicating $p < 0.0001$.

## Correlation between biomarkers and immune infiltration

To investigate the relationship between immune cells and key biomarkers, Figs. 4A–4B presented the distribution of different types of immune cells in normal and KIRC tissues. Overall, immune cells such as macrophages (M0, M1, M2) and T-cells (including CD8[+] T-cells, regulatory T-cells), and others were significantly more abundant in tumor tissues compared to normal tissues ($p < 0.05$), while B-cells (B cells naive) were more prevalent in normal tissues ($p < 0.05$). These differences showed slight variations between the TCGA training set and the ICGC validation set, for example, there was a significant difference in M2 macrophages between the tumor group and the control group in the TCGA training set, but not in the ICGC validation set. However, the overall results were consistent. Finally, the correlation between hub genes and immune cell scores in tumor samples was calculated and presented using correlation coefficients (ranging from −1 to 1) and significance levels (indicated by asterisks). The results indicated that these genes have significant correlations with multiple types of immune cells, and these correlations were particularly concentrated in T-cells and macrophages (Figs. 4C–4D). Among them, T cells CD4 memory activated

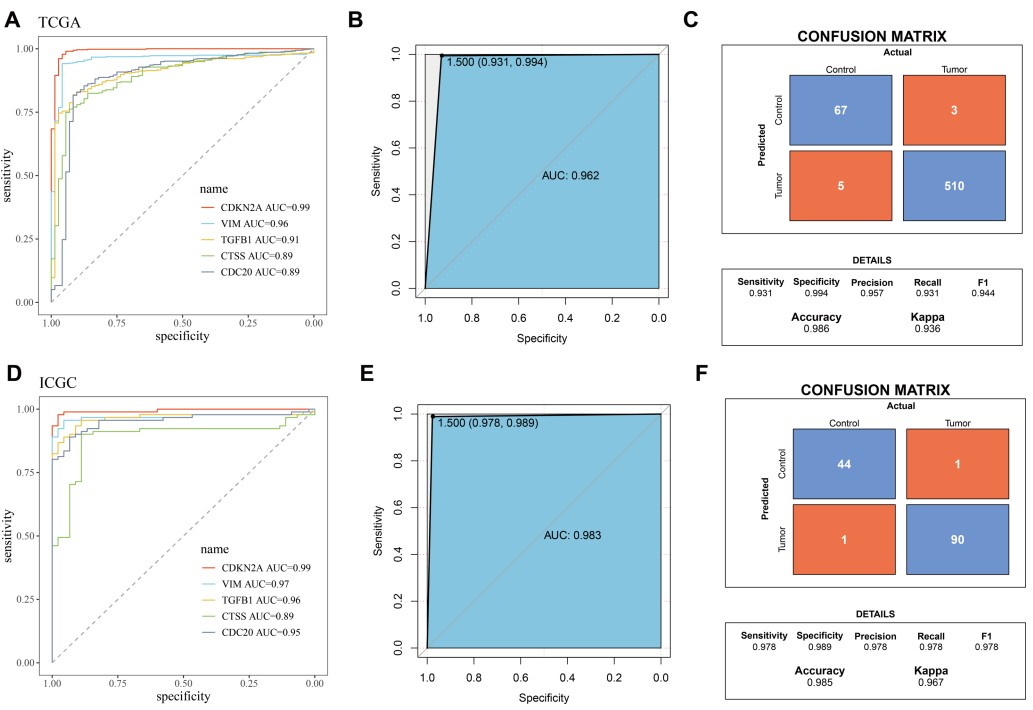

**Figure 3** **Validation of biomarkers and diagnostic model.** (A) ROC curves for hub genes' expression in TCGA; (B) ROC curve for overall predictive performance of the model in TCGA; (C) Confusion matrix showing classification results for tumor group and control group in TCGA; (D) ROC curves for hub genes' expression in ICGC; (E) ROC curve for overall predictive performance of the model in ICGC; (F) Confusion matrix showing classification results for tumor group and control group in ICGC.

and T cells CD4 memory resting showed significant differences across 5 biomarkers in the TCGA training set ($p < 0.05$).

## Relationship between hub genes and clinicopathological characteristics of KIRC

Next, we observed at the expression levels of the five hub genes in relation to clinical stage and grade. As shown in Fig. 5A, the expression levels of *CDKN2A* were significantly different in different stages and grades ($p < 0.01$). The expression of *VIM* and *CTSS* did not change significantly across different stages or grades (Figs. 5B, 5D). The trend in the expression level of the *TGFB1* gene across different stages and grades was not significant, although there was a slight increase in Stage III and IV, it did not reach statistical significance (Fig. 5C). The expression level of *CDC20* gene produced significant changes with the progression of clinical stage and grading, and its expression level was significantly up-regulated in Stage IV and G4 (Fig. 5E, $p < 0.001$). These results imply that these hub genes may be closely associated with tumor progression in KIRC.

## Distribution and expression of hub genes in the kidney

Six major cell clusters were identified from public scRNA-seq datasets. These clusters included endothelial cells, Natural killer T (NKT) cells, fibroblasts, B cells, tumor cells,

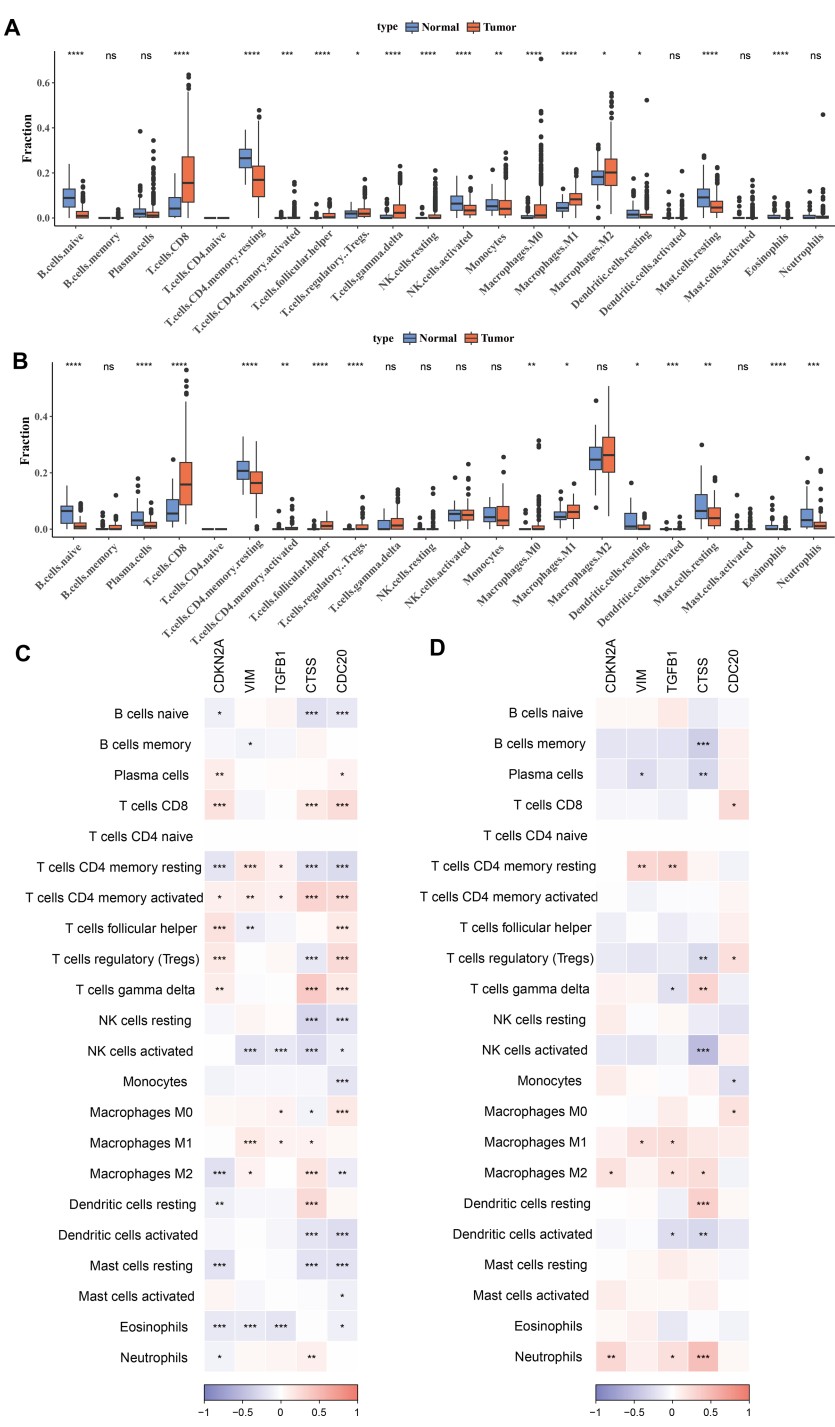

**Figure 4** **Correlation between biomarkers and immune infiltration.** (A) Distribution of different types of immune cells in normal and tumor tissues in the TCGA training set. (B) Distribution of different types of immune cells in normal and tumor tissues in the ICGC validation set. (C) Heatmap of correlations between biomarkers (CDKN2A, VIM, TGFB1, CTSS, CDC20) and different types of immune cells in the TCGA training set. (D) Heatmap of correlations between biomarkers (CDKN2A, VIM, TGFB1, CTSS, CDC20) and different types of immune cells in the ICGC validation set. * $p < 0.05$, ** $p < 0.01$, *** $p < 0.001$, **** $p < 0.0001$.

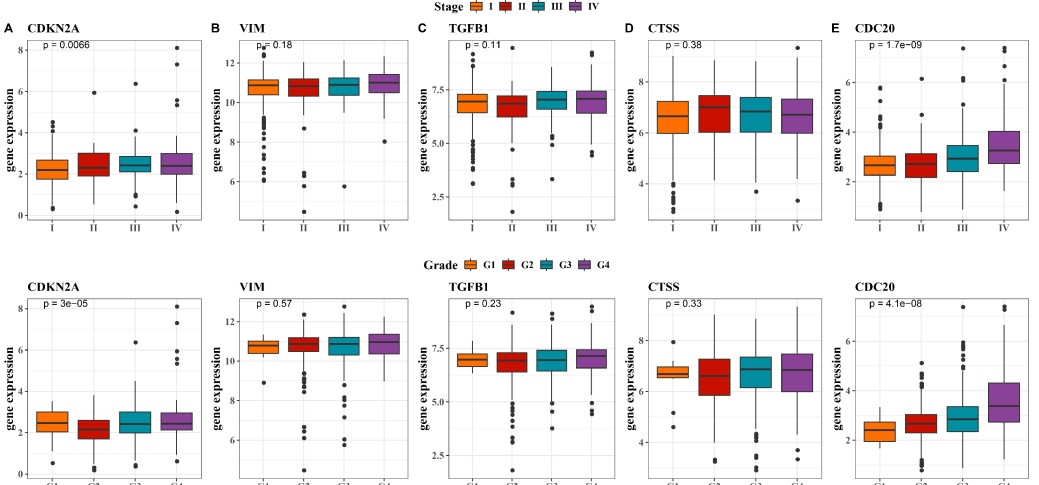

**Figure 5 Expression levels of 5 biomarkers (CDKN2A, VIM, TGFB1, CTSS, CDC20) in patients at different stages and grades.** (A) The expression levels of CDKN2A in patients with different stages and grades in the TCGA training set and the ICGC validation set; (B) The expression levels of VIM in patients with different stages and grades in the TCGA training set and the ICGC validation set; (C) The expression levels of TGFB1 in patients with different stages and grades in the TCGA training set and the ICGC validation set; (D) The expression levels of CTSS in patients with different stages and grades in the TCGA training set and the ICGC validation set; (E) The expression levels of CDC20 in patients with different stages and grades in the TCGA training set and the ICGC validation set.

and proliferative tumor cells (Fig. 6A). Further gene expression characteristics were used to demonstrate the marker genes of various cell types. For instance, genes highly expressed in B cells included *CD79A* and *MS4A1*, endothelial cells specifically expressed *PLVAP*, *VWF*, and *EMCN*, fibroblasts specifically expressed *COL3A1* and *COL1A2*, NKT cells specifically expressed *GZMA*, *NKG7*, and *GNLY*, while proliferative tumor cells and tumor cells expressed specific tumor marker genes such as *MKI67* and *EPCAM*, respectively (Fig. 6B). Comparison of the proportions of various cell types in the samples revealed that tumor cells accounted for the highest proportion, reaching 42%, followed by endothelial cells and B cells, each accounting for 19%, fibroblasts accounting for 17%, and NKT cells and proliferative tumor cells accounting for the smallest proportions (Fig. 6C). Finally, the expressions of key hub genes in different cell types was further analyzed. These results demonstrated that *TGFB1*, *CDKN2A*, and *CDC20* were significantly expressed in proliferative tumor cells, *CTSS* was most prominently expressed in B cells, while *VIM* was expressed in multiple cell types (Fig. 6D).

## Cellular validation based on *in vitro* experiment

The possibilities of *CDKN2A*, *VIM*, *TGFB1*, *CTSS* and *CDC20* as biomarkers of KIRC were further verified through *in vitro* experiments. The mRNA expression levels of these five genes were calculated, and it was found that the expressions of all these five genes in 786-O KIRC cells were significantly higher than those in 293T cells (Fig. 7A, $p < 0.05$). Since we observed a significant up-regulation of *CTSS* expression in KIRC cell lines relative to the other key genes screened, and its less studied in *CTSS* in KIRC. For this reason,
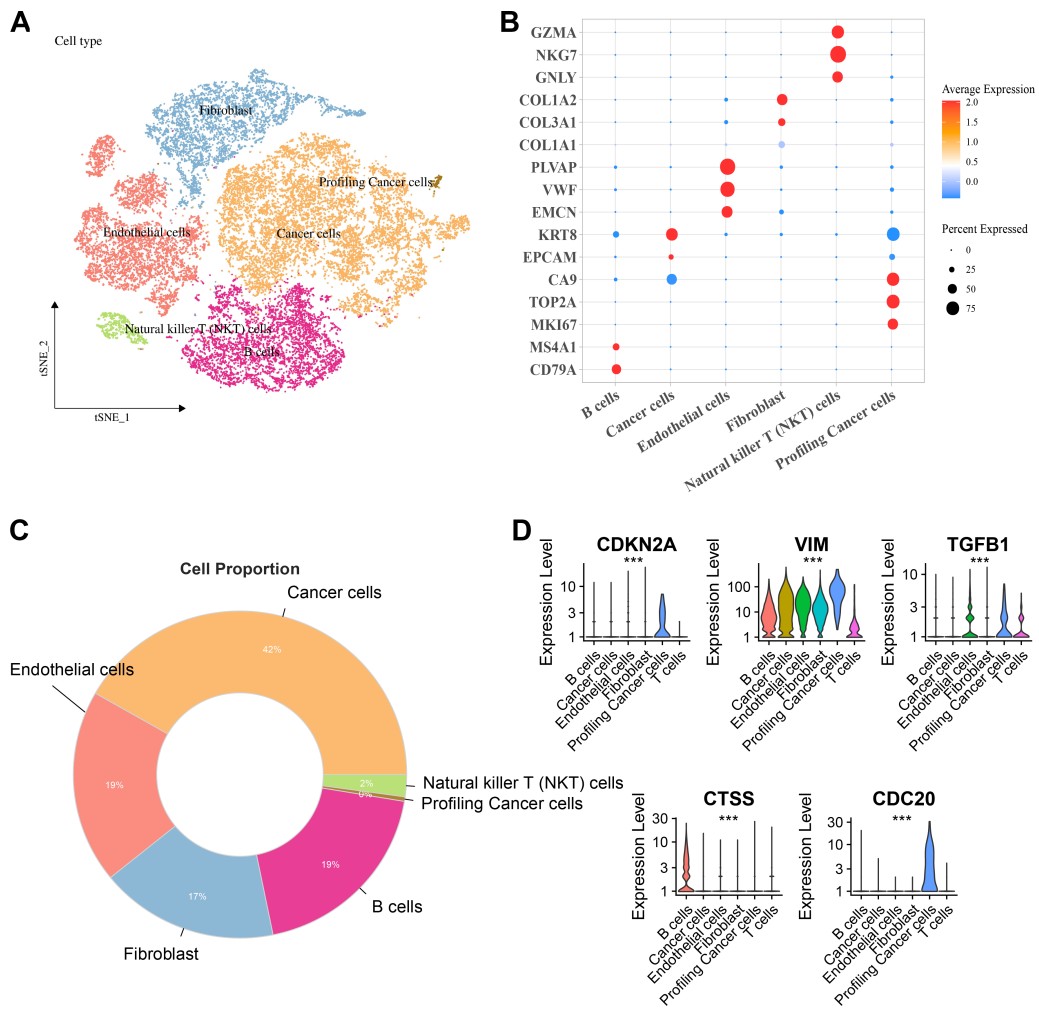

**Figure 6** **Distribution and expression of hub genes across six major cell clusters.** (A) Cluster analysis of KIRC cells using the TSNE dimensionality reduction method; (B) Bubble plot showing the expression levels of marker genes for each cell type; (C) Proportion distribution of various cell types across the entire KIRC sample; (D) Expression profile of hub genes in six major cell types. ***$p < 0.001$.

the impact of *CTSS* silencing on the biological function of KIRC cells was investigated (Fig. 7B). CCK-8 results showed that silencing the expression of *CTSS* significantly reduced the proliferative capacity of 786-O cells (Fig. 7C, $p < 0.001$). Furthermore, the relevant results demonstrated that the silencing of *CTSS* led to a reduction in the migration and invasion capabilities of KIRC cells (Figs. 7D–7E, $p < 0.001$).

# DISCUSSION

RCC accounts for about 2.2% of all newly diagnosed cancer cases, and over the past three decades, its incidence has been steadily rising across all stages (*Bray et al., 2024*). While early diagnosis of RCC is associated with a relatively favorable prognosis, KIRC is characterized by the absence of early warning signs (*Cuadros et al., 2013*). Notably, CICs

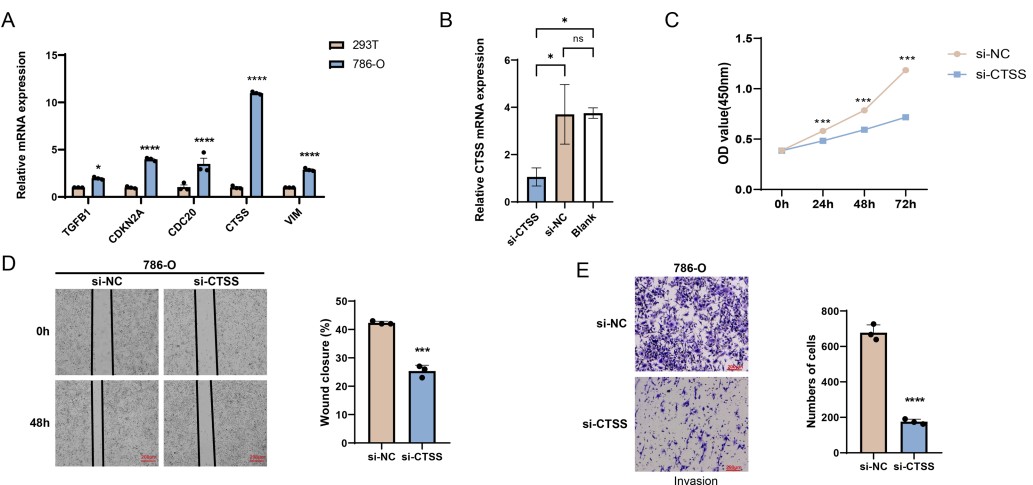

**Figure 7** ***In vitro*** **validation results.** (A) Quantified mRNA expression levels of *CDKN2A*, *VIM*, *TGFB1*, *CTSS*, and *CDC20* in 786-O cells and 293T cells. (B) qRT-PCR to verify *CTSS* knockdown efficiency in 786-O cells. (C) The effect of CTSS knockdown on 786-O cells proliferation was determined based on CCK-8. (D) The wound healing assay was used to assess the effect of *CTSS* knockdown on 786-O cell migration. (E) Transwell assay was used to assess the effect of *CTSS* knockdown on the invasive capacity of 786-O cells. All data of three independent trials were expressed as mean ± standard deviation. * $p < 0.05$, ** $p < 0.01$, **** $p < 0.0001$, and ns stands for no significant difference.

have been identified to occur between homotypic tumor cells or between immune cells and tumor (or other tissue cells) (*Mackay & Muller, 2019*), presenting a novel direction for KIRC research. Based on this, the current study employed machine learning methods to screen and identify five biomarkers (*CDKN2A*, *VIM*, *TGFB1*, *CTSS*, and *CDC20*). The reliability of these biomarkers has been verified by *in vitro* experiments. A diagnostic model with good predictive performance was established using these five biomarkers. In additional, the results of immune infiltration showed a higher proportion of T-cells and macrophages in tumor tissues. These findings open up new potential avenues for exploring and developing novel therapeutic approaches for KIRC, offering potential possibilities for improving patient treatment outcomes and prognosis.

Numerous studies have demonstrated that the PI3K-Akt signaling pathway exhibits aberrant activation during the genesis and development of various tumor types, KIRC included (*Makhov et al., 2018b*). Activation of this pathway typically results in dysregulated cell—cycle control, augmented anti-apoptotic capabilities, and immune evasion within the tumor microenvironment, all of which drive tumor progression and metastasis (*Xie et al., 2020*). Specifically in KIRC, the abnormal activation of the PI3K-Akt pathway is intricately associated with tumor cell proliferation, angiogenesis, and drug resistance (*Chen et al., 2023*). In our research, we observed a remarkable enrichment of upregulated genes from both the TCGA and ICGC datasets in this very pathway. Consequently, the PI3K—Akt pathway holds great potential as a viable therapeutic target for KIRC. Inhibiting its activity could be an effective strategy to curtail tumor growth and enhance the effectiveness of treatment regimens.

Importantly, this study identified five biomarkers, namely *CDKN2A*, *VIM*, *TGFB1*, *CTSS*, and *CDC20*. Among them, the *CDKN2A* gene is located in the frequently deleted p21 region on chromosome 9 and is widely recognized as a tumor suppressor (*Zhao et al., 2016*; *Gil & Peters, 2006*). The accumulation of various genetic alterations, including the *CDKN2A* gene, underlies the development of RCC (*Dulaimi et al., 2004*). *CDKN2A* is considered a key target for 9p deletions in multiple tumors, particularly RCC, due to its frequent inactivation through homozygous deletions or hypermethylation in the promoter region (*Vidaurreta et al., 2008*; *Schraml et al., 2001*). In 9.5% of RCC samples, the *CDKN2A* gene is lost along with other genetic materials, and these samples exhibit sarcomatoid features, a highly aggressive form of RCC that may benefit from immunotherapy (*Kiatprungvech et al., 2024*), which also corroborates with the significant correlation between the *CDKN2A* gene and immune cell infiltration observed in our study. Additionally, research has proven that inhibiting *CDKN2A* effectively promotes the formation of homologous CICs, and the activation of CIC-mediated cell death can serve as a barrier against potential malignant transformation induced by the inactivation of tumor suppressor genes like *CDKN2A* (*Liang et al., 2018*), suggesting that *CDKN2A*, as a CIC-related gene, holds potential therapeutic promise for KIRC and warrants further investigation. In this study, we observed an upward trend in *CDKN2A* gene expression levels in patients with KIRC, particularly those at G3 and G4 stages. However, contrary to this, earlier studies in laryngeal squamous cell carcinoma found an increased frequency of *CDKN2A* gene hypermethylation in patients at the G3 stage (*Smigiel et al., 2004*). This implies that the expression regulatory mechanisms of the *CDKN2A* gene may differ among different types of cancers, and there may be a complex relationship between its expression changes and cancer progression stages.

The *VIM* gene is located on chromosome 10p13 and serves as a major component of the mesenchymal cytoskeleton (*Shi et al., 2015*). Research on malignant tumors has shown that *VIM* functions crucially in cell cycle regulation, migration, adhesion, and the epithelial-mesenchymal transition (EMT) process in cancer (*Yao et al., 2020*). Recent research reports indicate that VIM protein can influence immune cells infiltration in the tumor microenvironment (*Dutsch-Wicherek, Lazar & Tomaszewska, 2011*). Prior studies have also found that *VIM* is an independent factor for the prognosis of KIRC (*Xu et al., 2020*). *TGFB1*, a cytokine with regulatory functions, has been reported in the literature to exhibit both stimulatory and inhibitory properties in regulating tissue homeostasis, developmental processes, tissue remodeling, and disease states such as cancer (*Ingman & Robertson, 2009*; *Ciftci et al., 2014*). Previous experiments have confirmed that the expression level of *TGFB1* is significantly elevated in KIRC tissues than normal kidney tissues (*Takahara et al., 2022*). *CTSS*, one of the 11 members of the cysteine protease family, is closely associated with various pathological conditions, including in cancers (*Wilkinson et al., 2019*). Recent research efforts have elucidated the key role of *CTSS* in influencing the pathogenesis of chronic kidney disease (*Steubl et al., 2017*). Experimental results show that the expression of *CTSS* is noticeably promoted in KIRC tissues compared to normal kidney tissues (*Zhou et al., 2024*). Based on this, in this study, *CTSS* was silenced for *in vitro* experimental verification, and the results also demonstrated that *CTSS* silencing could inhibit the migration and invasion of KIRC cells. Additionally, this study also found

that *CTSS* expression is particularly prominent in B cells. This finding echoed previous research, which also pointed out that *CTSS* exhibits high expression in antigen-presenting cells (APCs) and the lysosomes of malignant B cells (*Bararia et al., 2020*). In summary, these discoveries all confirm the correlation between *VIM*, *TGFB1*, and *CTSS* with KIRC, thus suggesting their potential as biomarkers for KIRC.

*CDC20* functions as an oncogenic regulator at multiple critical nodes of the cell cycle and is negatively regulated by the tumor suppressor protein p53, thus being considered a highly promising therapeutic target (*Wang et al., 2015*; *Kidokoro et al., 2008*). This study reveals that *CDC20* expression levels exhibit a significant increase with the advancement of clinical stage and grade. Notably, *CDC20* can directly bind to and activate the anaphase-promoting complex in conjunction with another important regulatory molecule, E-cadherin, which plays a vital part in the precise regulation of cell entry into and exit from mitosis (*Schrock et al., 2020*). Given that previous studies have shown substances such as APC$^{CDC20}$ to function in the transition from metaphase to anaphase by disrupting key cell cycle regulators (*Yu, 2007*), this discovery further implies that *CDC20* may occupy a pivotal position in disease progression (*Zeng et al., 2010*). Numerous studies have indicated that *CDC20* is not only a potential effective target for various cancer therapies but also a potential biomarker for prognosis (*Yuan et al., 2017*). Moreover, in previous research on KIRC, *CDC20* was also identified as a biomarker (*Gu et al., 2017*).

Our analysis of immune infiltration showed that the proportions of immune cells such as T cells and macrophages in tumor tissues were remarkably higher than those in normal tissues, and five biomarkers were identified to have significant correlations with T cells and macrophages. Previous studies have shown that in KIRC, the deletion or dysfunction of *CDKN2A* is closely associated with an inflammatory immune phenotype and the exhaustion state of CD8$^+$ T cells (*Sobottka et al., 2024*). Notably, the number of these exhausted CD8$^+$ T cells tend to increase relatively in metastatic sites, which may be linked to the immune escape mechanisms of tumors (*Sobottka et al., 2024*). On the other hand, *CTSS* within the cysteine cathepsin family is unique due to its limited tissue expression, primarily associated with antigen-presenting cells in lymph nodes and spleen, as well as other immune cells, especially macrophages (*Wilkinson et al., 2015*). Additionally, this study also found that the upregulated DEGs were enriched in immune regulation-related pathways such as T cell activation, Human T-cell leukemia virus 1 infection (HTLV-1), and regulation of T cell activation. Adult T-cell Leukemia/Lymphoma (ATL) is a CD4$^+$ T-cell malignancy caused by infection with HTLV-1 (*Liu et al., 2005*). *CDC20* plays an important part in the pathogenesis and development of ATL by mediating mitotic defects and the advancement of aneuploid cells (*Bruno et al., 2022*). This discovery also corroborates the correlation between the biomarkers obtained in this study and T-cell-related immune pathways. In summary, these findings could improve our understanding of the immune microenvironment in KIRC, but also offer potential targets for the development of novel immunotherapies targeting this disease.

However, this study has certain limitations. Firstly, although the data of this study underwent rigorous screening and analysis, it is constrained by the sample size and the homogeneity of data sources, which may, to a certain extent, undermine the broad

applicability and reliability of the research findings. To enhance the universality and persuasiveness of the conclusions, future research needs to expand the sample size and strive to encompass diverse patient populations and disease stages, while incorporating high-quality data from multiple sources. Furthermore, the specific molecular mechanism by which the biomarkers identified in this study affect immune infiltration remains unclear. Looking ahead, we plan to use advanced molecular biology techniques to deeply investigate the interactions between the identified biomarkers and various intracellular signaling pathways, with a particular focus on the specific pathways and mechanisms by which they act on immune infiltration, in the hope of revealing the underlying biological mysteries and providing a more solid theoretical basis for the diagnosis of KIRC and treatment.

## CONCLUSION

This study identified five core biomarkers associated with CICs in KIRC through transcriptome analysis and machine learning methods: *CDKN2A*, *VIM*, *TGFB1*, *CTSS*, and *CDC20*. The diagnostic model constructed based on these biomarkers demonstrated good predictive performance. Importantly, these biomarkers were significantly correlated with the infiltration of specific immune cells in the tumor microenvironment, suggesting that these genes may be involved in regulating the immune evasion mechanism of KIRC. Additionally, the expression levels of CDKN2A and CDC20 showed significant differences in clinical stage and pathological grade. Finally, *in vitro* experiments verified that silencing CTSS inhibited KIRC cell proliferation, migration, and invasion, further supporting its potential as a therapeutic target. In conclusion, this study provides new molecular targets for the early diagnosis, prognosis assessment, and personalized treatment of KIRC, offering important insights for future therapeutic strategies.

### Abbreviations

| | |
|---|---|
| **RCC** | renal cell carcinoma |
| **CIC** | cell-in-cell structure |
| **KIRC** | kidney renal clear cell carcinoma |
| **TCGA** | The Cancer Genome Atlas |
| **GDC** | Genomic Data Commons |
| **API** | Application Programming Interface |
| **FPKM** | Fragments Per Kilobase of exon model per Million mapped fragments |
| **TPM** | Transcripts Per Million |
| **RECA-EU** | The Renal Cell Cancer-European Union |
| **ICGC** | International Cancer Genome Consortium |
| **GEO** | Gene Expression Omnibus |
| **DEG** | differentially expressed gene |
| **GO** | Gene Ontology |
| **KEGG** | Kyoto Encyclopedia of Genes and Genomes |
| **BP** | biological processes |
| **CC** | cellular components |
| **MF** | molecular functions |

| | |
|---|---|
| **RFE** | Recursive Feature Elimination |
| **CV** | cross-validation |
| **SVM** | Support Vector Machine |
| **LASSO** | Least Absolute Shrinkage and Selection Operator |
| **ROC** | Receiver Operating Characteristic |
| **AUC** | Area Under the Curve |
| **scRNA-seq** | single-cell RNA sequencing |
| **PCA** | Principal Component Analysis |
| **TSNE** | t-Distributed Stochastic Neighbor Embedding |
| **DMEM** | Dulbecco's Modified Eagle Medium |
| **RPMI** | Roswell Park Memorial Institute |
| **FBS** | fetal bovine serum |
| **qRT-PCR** | quantitative real-time PCR |
| **NCBI** | National Center for Biotechnology Information |
| **CAM** | Cell adhesion molecule |
| **NKT cell** | Natural killer T-cell |
| **EMT** | epithelial-mesenchymal transition |
| **APC** | antigen-presenting cell |
| **ATL** | Adult T-cell Leukemia/Lymphoma |
| **HTLV-1** | Human T-cell Leukemia Virus Type 1 |

### Funding

The authors received no funding for this work.

### Competing Interests

The authors declare there are no competing interests.

### Author Contributions

- Zehua Wang conceived and designed the experiments, performed the experiments, analyzed the data, prepared figures and/or tables, authored or reviewed drafts of the article, and approved the final draft.
- Zhongxiao Zhang conceived and designed the experiments, performed the experiments, analyzed the data, prepared figures and/or tables, authored or reviewed drafts of the article, and approved the final draft.

### Data Availability

The sequences are available at GEO: GSE224630.

The raw data is available in GitHub and Zenodo:

- https://github.com/zhongxiaoZhang/Raw-data.git

- zhongxiaoZhang. (2025). zhongxiaoZhang/Raw-data: raw data (v.1.1.1). Zenodo. https://doi.org/10.5281/zenodo.14922228.

## Supplemental Information

Supplemental information for this article can be found online at http://dx.doi.org/10.7717/peerj.19246#supplemental-information.

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
