# Peer review of "Biomarkers associated with cell-in-cell structure in kidney renal clear cell carcinoma based on transcriptome sequencing"

_PeerJ, doi:10.7717/peerj.19246_

## Round 0.1 · original submission · Major Revisions

Based on the evaluations from two reviewers, who recommended major and minor revisions respectively, I have carefully assessed their feedback. Given the scope and nature of the concerns raised, a major revision is required. Please address all comments from both reviewers thoroughly, with special attention to Reviewer 1's major concerns. A detailed point-by-point response to all reviewer comments should accompany your revised manuscript.

Reviewer 1 ·

Basic reporting

1. The first occurrence of an abbreviation in the abstract should be presented in the format "Full Name (Abbreviation)", and thereafter, the abbreviation should be used, such as KIRC, CIC, etc.

2. What is the method for determining migration rate in a 2.7 Cell migration assay?

3. By stating "assuming they followed a normal distribution," does the author mean that no normality test was conducted, and it's merely an assumption?

4. What does the p-value in Fig.5 signify, and how is it concluded that G3 and G4 patients express 232 significantly higher than G2 patients in Fig. 5a? It seems there is a discrepancy between the text description and the statistical results. Similarly, there is the same issue with Fig. 5e.

5. What is the specific reason for selecting CTSS for further experiments in Fig.7?

6. Fig.7 requires additional evidence of successful CTSS knockdown, at least using qPCR results to show that the expression of CTSS in the si-CTSS group is lower than in the si-nc group, and it is necessary to introduce the grouping and the meaning of each group in the methodology.

7. At least one proliferation assay should be added in Fig.7 to demonstrate the changes in cell proliferation after CTSS knockdown.

8. Fig.7d,e The images are too blurry, with indistinct cell morphology and unclear scale bars.

9. The conclusion should encompass all experimental findings, including in vitro validation experiments.

10. Fig.6d A comparison has been made, and it is necessary to add significance symbols to express that the statistical comparison results are significant.

11. The figure legend for Fig.7 needs to indicate the sample size.

12. The image resolution is not high; please provide high-definition images.

Experimental design

no comment

Validity of the findings

no comment

Reviewer 2 ·

Basic reporting

The present research discovers 5 biomarkers related to cell-in-cell structure (CICs) in kidney renal clear cell carcinoma through transcriptome sequencing and in vitro experiments, offering valuable insights into its developmental mechanism for the research of CICs in this disease. Whereas, the following problems need to be explained before publication.
In the Abstract, the full name of abbreviations “KIRC” and “CIC” is absent, which is required to be supplemented into the manuscript. Meanwhile, it is needed to check other parts of the text, such as “BP, CC, MF” (Line 180).
In the methods of Abstract section, Line 28-34, the description of single-cell data processing is too tedious, only the key methods or packages are need to state in the Abstract. Please redescribe this section briefly.
In the preface, the author introduces the effects of cell-in-cell structures (CICs) on tumor cells, but is there any relevant research reports on the involvement of CICs in KIRC? It should add more about the progress of CICs in KIRC and point out the current research challenge or research gap to highlight the significance of this study.
How many KIRC samples of the single-cell dataset GSE224630 contains, which needs to be clarified in the manuscript.
In the method part of identification and enrichment analysis of DEGs, why did the authors identify only the common up-regulated genes? Could the author simply explain the reasons?
The subheading of section 2.5 is seemingly not appropriate; it is suggested to change it to a more suitable title.
The involvement of 5 biomarkers in the progression of KIRC is elaborated at length in the discussion part, I wonder if the high expression levels of these genes in KIRC are associated with patient outcomes.
Functional enrichment analysis displays that the upregulated DEGs are not only enriched in T cell activation and Human T-cell leukemia virus 1 infection, but also significantly implicated in the PI3K-Akt signaling pathway and Cytokine-cytokine receptor interaction. PI3K-Akt is a key signaling pathway within cells that plays a vital role in a variety of biological processes. Thus, the potential effects of these pathways in KIRC development are expected to extensively discuss.

Experimental design

no comment

Validity of the findings

no comment

---

## Round 0.2 · accepted · Accept

I have carefully reviewed your revised version along with your responses to the reviewers' comments. Both reviewers have found that all their concerns have been satisfactorily addressed and have recommended acceptance of your manuscript. Based on these positive recommendations, I am pleased to inform you that your manuscript has been accepted for publication.

Reviewer 1 ·

Basic reporting

Thank you for the re-invitation from the editor. I have carefully read the manuscript, and the author has provided detailed responses to my comments, effectively addressing the main issues. I believe it is ready for publication in its current form.

Experimental design

no comment

Validity of the findings

no comment

Reviewer 2 ·

Basic reporting

The present research discovers 5 biomarkers related to cell-in-cell structure (CICs) in kidney renal clear cell carcinoma through transcriptome sequencing and in vitro experiments, offering valuable insights into its developmental mechanism for the research of CICs in this disease.

Experimental design

no comment

Validity of the findings

no comment